# Miresga: Accelerating Layer-7 Load Balancing with Programmable Switches

## Abstract

As online cloud services expand rapidly, layer-7 load balancing has become indispensable for maintaining service availability and performance. The emergence of programmable switches with both high performance and a certain degree of flexibility has made it possible to apply programmable switches to load balancing. Nevertheless, the meager memory capacity and the relatively sluggish speed of table entry insertion and deletion of programmable switches have severely constrained their performance.

To this end, we introduce Miresga, a hybrid and high-performance layer-7 load balancing system by co-designing hardware and software. The core idea of Miresga is to maximize the utilization of hardware and software resources by rationally partitioning the layer-7 load balancing task, thereby improving performance. To achieve this, Miresga offloads the elephant flows, which account for the majority of traffic, to programmable switches that excel at packet processing, and Miresga utilizes general-purpose servers with stronger computational capabilities to parse application layer protocols and apply load balancing rules. To alleviate memory pressure on the programmable switch, Miresga employs a back-end agent to handle memory-intensive tasks, working in conjunction with the programmable switch to complete the offloaded tasks. This design leverages the performance advantages of the programmable switch while avoiding bottlenecks caused by its limited memory and table insertion speed. We implement the Miresga prototype with a 3.2 Tbps Intel Tofino switch and general-purpose servers. The evaluation results show that Miresga achieves 3.9× throughput and 0.4× latency compared to software load balancing solutions. Compared to state-of-the-art design employing programmable switches, Miresga achieves almost the same throughput and latency for delivering large objects and 5.0× throughput and 0.2× latency when transmitting small objects.

## 1 Introduction

Modern online service providers have utilized load balancing in cloud data centers to distribute traffic across large server clusters [5]. An online service usually receives traffic from outside the data center through one or a few virtual IPs (VIPs), and each server in the cluster is assigned an individual direct IP (DIP) address. The load balancers (LBs) distribute the traffic destined for VIPs among the servers and route it to the specified servers based on pre-configured rules. However, packet-level header checking and port selection of LBs introduce significant overhead in both throughput and latency. Therefore, the performance of the LB has a substantial impact on the quality of service.

As the complexity of online services continues to increase, layer-4 (L4) load balancing only using the information of the IP/TCP header no longer fully meets the requirements. Layer-7 (L7) load balancing allows LBs to determine the destination servers based on the fields in the application layer contents like domain names or URI paths [5], offering more fine-grained and more secure load balancing.

Traditional L7 load balancing solutions predominantly fall into two categories: software-based LBs, which operate on commercial servers, and hardware-based LBs, which run on fixed-function application-specific integrated circuits (ASICs). Software-based LBs [3, 6, 9] are limited by the packet processing speed of CPU architecture and the bandwidth of network interface cards (NICs), necessitating extensive server scaling to handle high traffic volumes and many concurrent connections. Moreover, there is competition for limited CPU resources among different flows. The presence of *elephant flows*—where a small number of flows dominate the traffic—can lead the load balancer to prioritize these, causing increased latency for other flows [34]. In contrast, hardware-based LBs [1, 4] deliver high performance on individual machines but are expensive and challenging to scale. Most hardware solutions rely on DNS-based load balancing, which can only distribute traffic by domain name and lacks support for other protocols [4].

The emergence of programmable network hardware, *e.g.*, the programmable switch, has enabled offloading some stateless operations onto the hardware, thereby significantly reducing the load on server CPUs. The programmable switch leverages its programmable ASIC to achieve ~Tbps line-rate packet processing and allows operators to modify the header based on customized rules. So by substituting programmable switches for ToR switches, like Prism [35], we can offload plenty of traffic from servers, reducing the number of servers and saving cost. However, the extremely limited memory of programmable switches and the disparity between the speed of packet processing and the rate of table entry update have emerged as bottlenecks, hindering the full exploitation of the performance advantages offered by programmable switches.

To this end, we propose a hybrid and high-performance L7 load balancing solution, Miresga, to accelerate L7 load balancing by co-designing hardware and software. The reason for the co-design of software and hardware is that programmable switches can achieve high-speed forwarding and high throughput. Still, memory resources and L7 protocol processing capabilities are limited. On the contrary, general-purpose servers have rich memory resources and strong L7 protocol processing capabilities but poor forwarding performance. Therefore, the combination of the two is expected to achieve better performance. The core idea of Miresga is to improve the performance of L7 load balancing by strategically dividing the task to maximize the use of software and hardware resources and capabilities. Through careful observation, we divide L7 load balancing into three distinct tasks: 1) establishing connections with clients and servers respectively, 2) passing application layer protocols and applying load balancing rules, and 3) subsequent packet forwarding through splicing connections. The L4 connection establishment with the client does not involve the application layer protocol parsing in task 1); thus, we can use programmable switches to handle it.

In contrast, task 2) is more suitable for implementation on general-purpose servers (referred to as *front-end servers* in our design). Regarding task 3), while it can be managed by the programmable switch alone, the potential bottleneck arises from the limited speed at which the local control plane of the programmable switch can push updates to its data plane. Therefore, Miresga adopts the parallel software and hardware strategy to process task 3) with two paths. The fast path through the programmable switch is reserved for handling *elephant flows*—large data transfers—while the remaining *mouse flows*—small data transfers—are processed via the slow path that involves the participation of general-purpose servers.

However, realizing this idea still faces three challenges. First, for the flows offloaded to the programmable switch, the programmable switch needs to store the information of these to perform connection splicing correctly. However, storing the complete information for these flows becomes daunting due to the limited SRAM memory in the programmable switch. Second, since the L7 load balancer must establish connections with both the client and the back-end server, using a complex TCP kernel stack can severely impact performance and consume a significant amount of resources. Third, given that Miresga is a hybrid system and TCP is a stateful protocol, it is crucial to synchronize the flow state between the hardware and software components to guarantee the reliability of data transmission. To solve the challenges mentioned earlier, Miresga proposes the following three designs:

- **Efficient Connection Splicing (§4.1)**: Miresga employs two approaches to alleviate memory pressure on programmable switches while ensuring the correct connection splicing. Miresga further decomposes task 3) into the modification of IP and port and the synchronization of sequence numbers and acknowledgment numbers. For the former, we save space by compressing the table entries; as for the latter, since sequence number and acknowledgment number synchronization require recording the initial sequence number differences, which can not be compressed, the programmable switch is no longer responsible for this part. Instead, we design a *back-end agent* to handle this memory-intensive task. The back-end agent uses eBPF [25] programs in front of the kernel protocol stack to modify the sequence number or the acknowledgment number of the packets in advance, thus ensuring the Web server can process the requests normally.

- **Lightweight Protocol Stack (§4.2)**: To skip the kernel protocol stack, we design a lightweight protocol stack that merges the states of the connections the LB established with both the client and the server into a single state of this flow, consuming far fewer resources. At the same time, the front-end server does not need to perform the congestion control, lightening the burden to some extent.

- **State Consistency Maintenance (§4.3)**: In Miresga, two types of states require synchronization: 1) the initial sequence numbers that the client, the programmable switch, and the Web server choose, and 2) the information for splicing connections to the offloaded flows. To synchronize the initial sequence number, Miresga fuses the initial sequence numbers into the regular packets to avoid additional delivery. For the latter one, the extra transmission is essential. Miresga does not require strict state

consistency between the passing parties but allows a short out-of-sync period. Instead of blocking traffic, traffic is handled by the front-end server temporarily.

We implement a prototype of the Miresga programmable switch on a 3.2 Tbps Intel Tofino [7] switch and both the front-end server and back-end agent on the general-purpose server (§5). Our experimental results (§6) prove that Miresga can achieve $2.0 \sim 3.9\times$ throughput with a 40% latency compared to HAProxy [6] with DPDK [36] accelerating. Compared to state-of-the-art programmable switch design Prism [35], Miresga achieves almost the same throughput and latency for delivering large objects and $5.0\times$ throughput and $0.2\times$ latency when transmitting small objects. Our experiments also demonstrate that the presence of elephant flows in the traffic does not affect the performance of our load balancing due to offloading the elephant flows to the programmable switch with high throughput. In contrast, the latency of software load balancing will increase significantly. Our prototype implementation is available at https://github.com/Miresga-L7LB/Miresga.

## 2 Background and Related Work

In this section, we first offer a brief background of L7 load balancing. Then, we briefly introduce the programmable switch and related work.

### 2.1 Background

L7 load balancing uses application layer data for load balancing. Currently, the majority of L7 load balancer designs follow a proxy-like architecture. In this setup, the load balancer must first establish a connection with the client and parse the application layer protocol to determine the appropriate back-end server to handle the request. Unlike L4 load balancers, L7 load balancers must establish distinct connections to both the client and the back-end server. This process introduces additional complexity and can lead to a decrease in performance. Moreover, because a persistent connection may contain multiple requests, the same back-end server will not be guaranteed to process each one. This necessitates the load balancer to recognize new client requests, terminate the ongoing connection, and establish a fresh connection with an alternative back-end server as the situation demands.

### 2.2 Related Work

**Programmable Switches.** Similar to traditional switching devices, programmable switches offer extremely high throughput (~Tbps). While maintaining high performance, programmable switches allow operators to customize packet processing logic through domain-specific languages like P4 [17], including customized header parsing and modification. The packet processing pipelines (*i.e.*, the data plane running on the ASIC) of programmable switches consist of a sequence of reconfigurable match-action tables (RMTs) [18] stored in the in-switch SRAM. Operators can manage table entries through the built-in CPUs (*i.e.*, the local control plane) on the programmable switches. Furthermore, the programmable switch can send parts of a packet or the entire packet to the CPU via the digest function or specific PCIe channels. Due to the combination of programmability and high performance, programmable switches have been applied in many fields, such as congestion control [11, 20, 44],

cloud gateways [57], security [38, 70, 72], and in-network computing [37, 42, 62, 68]. There is also some work focusing on the L4 load balancing [30, 52, 71], but programmable switches struggle to parse variable-length headers of the application layer, necessitating a redesign to address this issue.

**Software L7 Load Balancers.** HAProxy [6] and Nginx [9] use the kernel stack, establishing connections with both the client and the server. This design requires the LBs to maintain the state of both connections simultaneously and use TCP splicing [49] to transfer received data between the two connections. These LBs need to perform congestion control, which consumes huge CPU resources. Yoda [29] attempts to solve these problems by using tunneling operations to adjust the IP/TCP headers to splice the two connections so that the congestion control will be performed by the server and the client. However, tunneling operations and storing information on other servers introduce additional latency. In addition, the receiving side scaling (RSS) is widely used in commercial NIC. The NIC attempts to distribute the flows evenly among the CPU cores, and one CPU core handles the entire processing of the flow. However, this per-flow allocation may lead to multiple elephant flows processed on a CPU core, and the task of this CPU core is extra heavy, which affects the transmission efficiency of other flows on this core [34].

**L7 Load Balancers Accelerated with Programmable Hardware.** Prism [35] uses a set of front-end servers in combination with a programmable switch to complete load balancing. The front-end server passes the serialized TCP connection so that the back-end server can easily reestablish the connection from it. The IP/port is modified with the programmable switch, and the subsequent packet delivery is directly sent through the programmable switch without the front-end involvement. However, the design that needs to frequently publish entries to the programmable switch is limited by the limited modification speed of the programmable switch, which introduces additional delay. AccelTCP [53] leverages the SmartNIC to accelerate the TCP stack by offloading the connection setup and teardown to the SmartNIC so that the CPU can concentrate on the processing of subsequent packets. It also supports offloading the TCP splicing to the SmartNIC, but the CPU will not be able to detect any more packets for this flow. It is not suitable for L7 load balancing because some flows may send multiple requests that need to be processed by different servers, which asks the LB to close the old connection and establish a new connection. Moreover, the cost of equipping LBs with SmartNICs is huge.

## 3 Miresga Overview

Before delving into the design details of Miresga, we first provide its high-level overview. This section begins with the architecture of Miresga and then follows with an example demonstrating its workflow.

### 3.1 Architecture

Given the current limitations of both software-based and hardware-accelerated load balancing solutions, Miresga seeks to bridge the gap by combining the flexibility of CPU-based header parsing and state-of-the-art software optimization techniques with the high-speed packet processing capabilities of programmable ASICs to enhance L7 load balancing. Miresga divides the L7 load balancing

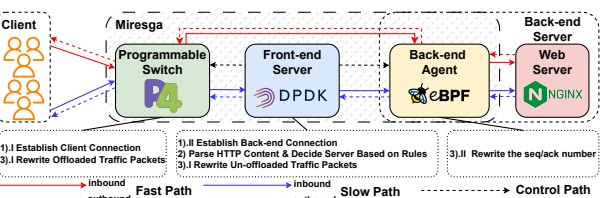

**Figure 1: Miresga architecture.**

task into three main parts: 1) connection establishment, 2) application layer protocol parsing, and 3) subsequent packet forwarding through splicing two connections. Figure 1 shows the architecture of Miresga, which mainly consists of three components: the programmable switch, the front-end server, and the back-end agent running on the back-end server. We use these three components together to complete three tasks:

**1) Programmable Switch.** The programmable switch not only needs to fulfill its original forwarding function but also takes on some simple stateless tasks. We assign it the task of establishing connections with clients, i.e., returning a SYN-ACK packet when receiving a SYN packet from the client. Considering that the design of the programmable switch is better suited for handling elephant flows, we also delegate one part of the task of connection splicing on elephant flows after the two connections have been established. The programmable switch retrieves elephant flow information from its local table and then modifies the IP and port based on this information.

**2) Front-end Server.** With the flexible CPU, the front-end server is responsible for parsing the application layer protocol, determining whether the flow qualifies as an elephant flow, and selecting the back-end server based on the pre-configured rules. Then the front-end server establishes the connection with the back-end server. After the connection is established, the front-end server will send the information of the flow to the programmable switch if the flow is an elephant flow. Otherwise, the front-end server will perform the task like the programmable switch, modifying the IP and port.

**3) Back-end Agent.** Due to the limited resources of the programmable switch, we delegate the resource-intensive task of sequence number and acknowledgment number synchronization to the back-end agent, which has more abundant memory. The back-end agent modifies the sequence or acknowledgment numbers to ensure that the sequence and acknowledgment numbers received by the client and Web server match those stored on each side.

### 3.2 Workflow by Example

In this subsection, we demonstrate an example of Miresga processing an HTTP 1.1 GET request to show its workflow. Figure 2 shows how Miresga works on a flow. When a client starts to establish a connection (①), the programmable switch just returns a SYN-ACK packet with a random acknowledgment number (②), awaiting the establishment of the connection and the client sending an HTTP request. Upon receiving the HTTP request (③), the programmable switch transmits it to the front-end server (④) to process. The front-end server caches this packet, parses the relevant parts of the HTTP request, determines the back-end server based on pre-configured rules, and then sends a special SYN packet with the acknowledgment number that was replied by the programmable switch to the selected server to establish a connection (⑤). After receiving the

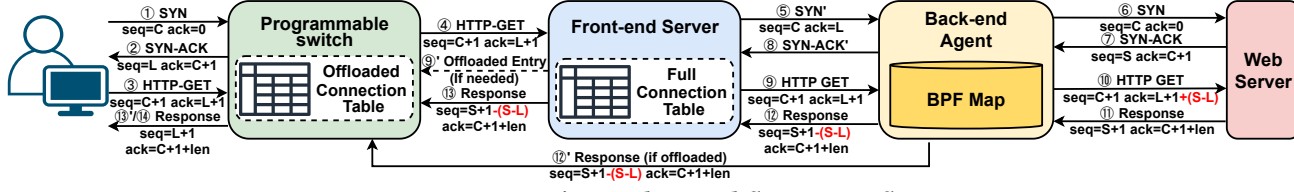

Figure 2: Miresga's complete workflow on one flow.

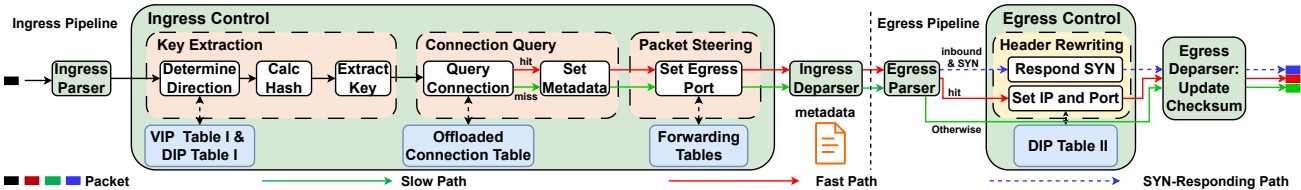

Figure 3: The workflow of Miresga Programmable switch

SYN packet, the back-end agent will store the acknowledgment number to the BPF Map with pre-allocated memory specifically for storing sequence number information and then send a normal SYN packet to the kernel stack (⑥), waiting for the kernel stack to return the SYN-ACK packet (⑦). It will calculate the difference, store it in the BPF Map, and send this to the front-end server (⑧). The front-end server then sends the cached packet to the back-end server (⑨). If the front-end server finds this flow is an elephant flow according to the L7 protocol, the front-end server will send the entry to the programmable switch (⑨'). The local control plane will write this entry to the Offloaded Connection Tanle. For the subsequent responses, the programmable switch or the front-end server will modify the IP and port. The back-end agent will modify the sequence and acknowledgment number. For the inbound traffic packets, Miresga will modify its $\langle dstIP, dstPort \rangle$ to the real IP and port ($\langle DIP, DPort \rangle$) of the back-end server, and for the outbound traffic packets, Miresga will modify its $\langle srcIP, srcPort \rangle$ to $\langle VIP, VPort \rangle$. So, the client can not get the real IP and port of the back-end servers, thus protecting them to a certain extent. When receiving FIN/RST packets from the client or the back-end server, we will remove the entry from the Offloaded Connection Table, Full Connection Table, and the BPF Map.

## 4 Design Details

This section presents the design details of Miresga. We mainly introduce our designs on how to solve the three challenges mentioned before.

### 4.1 Efficient Connection Splicing

Connection splicing is carried out jointly by three components. Once the connection with the back-end server is established, the flow enters the connection splicing phase. Depending on whether the flow is offloaded to the programmable switch, the IP/port is modified either by the front-end server (i.e., slow path) or the programmable switch (i.e. fast path). The back-end agent is responsible for modifying the sequence and acknowledgment numbers before the Web server processes the request. In this way, two connections are merged into one. We will explain how the front-end server performs in §4.2. In this subsection, we mainly introduce the design of the programmable switch and the back-end agent.

*4.1.1 Programmable Switch in Action.* This sub-section describes how programmable switches handle connection splicing for offloaded flows and correctly forward the remaining flows. In Miresga, the programmable switch mainly has five key components:

- **Key Extraction**: This module first determines the direction of this flow and then computes the key corresponding to the direction.
- **Connection Query**: It occupies the majority of stages in the ingress pipeline. Based on the computed key, we find the corresponding entry in the Offloaded Connection Table and set up the metadata to pass it to the egress pipeline.
- **Packet Steering**: This module determines which egress port the packet should be forwarded to according to whether the packets belong to the offloaded traffic and according to the flow direction.
- **Header Rewriting**: It is responsible for adjusting the IP and port based on the metadata set by the Connection Query module. It also needs to generate the SYN-ACK packet to establish a connection with the client.
- **Offloaded Connection Table**: It stores the compressed information of the offloaded connections and can be accessed by the Connection Query module. The control plane can also use the vendor-provided southbound APIs to update its entries.

**Workflow.** Figure 3 shows the workflow of the data plane. All packets need to be processed in the ingress pipeline first and then in the egress pipeline. Miresga comprises four modules in the ingress pipeline. First, packets are processed in the Key Extraction module to extract the key and determine the direction of the packet. After obtaining the key, the Connection Query module checks if the key exists in the Offloaded Connection Table. Then Miresga sets the compressed information stored in the Offloaded Connection Table as metadata. The Packet Steering module determines the egress port of the packet based on its direction and whether it belongs to the offloaded flow. In the egress pipeline, the primary task is to perform connection splicing based on the metadata. Only packets belonging to the offloaded flows will be rewritten. The Header Rewriting module will modify the IP and port based on the metadata. Especially, for the inbound SYN packet, the Header Rewriting module will return a SYN-ACK packet.

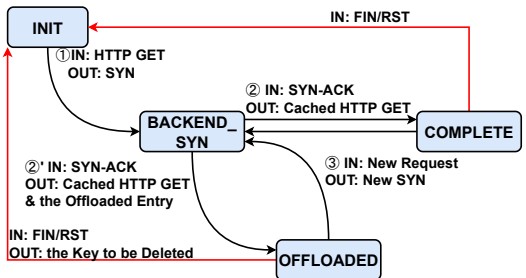

**Figure 4: The state transition diagram of the TCP FSM**

**Entry Compression.** Since programmable switches have very limited memory resources, the capacity would be severely restricted if we stored complete information about each connection. Hence, Miresga saves memory space by compressing table entries. Considering that using hash values as keys requires the involvement of the programmable switch's slow control plane to handle hash collisions, Miresga adopts an alternative method to compress the table entries.

For both inbound and outbound packets of the same flow, the common point is that either the $\langle srcIP, srcPort \rangle$ or the $\langle dstIP, dstPort \rangle$ matches the $\langle CIP, CPort \rangle$. Also, as each TCP connection is uniquely identified by its $\langle srcIP, srcPort \rangle$ and $\langle dstIP, dstPort \rangle$ tuple, there can only be one flow between the same $\langle CIP, CPort \rangle$ and $\langle VIP, VPort \rangle$ at any given time. Miresga assumes that a programmable switch will be set to just one $\langle VIP, VPort \rangle$, so at the same time, $\langle CIP, CPort \rangle$ can only correspond to one flow. Hence we can use the tuple $\langle CIP, CPort \rangle$ as the identifier for this flow. Since both $\langle VIP, VPort \rangle$ and $\langle DIP, DPort \rangle$ are set up by the service provider, we can access all $\langle VIP, VPort \rangle$ and $\langle DIP, DPort \rangle$ tuples that the programmable switch needs to handle and store them in the programmable switch in two tables called VIP Table and DIP Table I. In the entries of the Offloaded Connection Table, we can use one index $D\_index$ to compress $\langle DIP, DPort \rangle$ tuples and store the mapping of $D\_index$ and $\langle DIP, DPort \rangle$ in another table called DIP Table II. Hence, the Key Extraction module can determine whether the $\langle srcIP, srcPort \rangle$ is in the DIP Table I and whether the $\langle dstIP, dstPort \rangle$ is in the VIP Table. If the VIP Table is hit, the packet is considered inbound, and $\langle CIP, CPort \rangle$ represents $\langle srcIP, srcPort \rangle$. If the DIP Table I is hit, the packet is considered outbound, and $\langle CIP, CPort \rangle$ represents the $\langle dstIP, dstPort \rangle$.

Another task is how to schedule these modules within a limited number of stages. The pipelined design of a programmable switch means that each of our modules takes up a certain number of stages, but we can combine them to reduce the number of stages by placing them in ingress and egress pipelines respectively. Through this, the other modules are placed in as less stages as possible, so that the Offloaded Connection Table has more stages. The detail of the table arrangement is described in Appendix A.

*4.1.2 Back-end Agent in Action.* The programmable switch can complete the modification of IP and port, as well as the synchronization of sequence numbers and acknowledgment numbers entirely on its own. However, unlike IP and port, which can be compressed without loss, it is challenging to compress the nearly random difference between sequence numbers and acknowledgment numbers. This necessitates storing the full 32-bit difference in the limited

memory of the programmable switch. Expanding the size of the entry would further increase the bandwidth consumed by the operation of offloading the entry, thereby impacting system performance. Given that the server has ample memory space compared to the limited memory of the programmable switch, we opt to introduce a *back-end agent* to handle the more memory-intensive task of sequence number and acknowledgment number synchronization. We use eBPF to implement the back-end agent. Since the eBPF program runs in the kernel and the operations we perform are simple, the back-end agent has a negligible impact on throughput or latency. The back-end agent consists of two main parts: Ingress Program using XDP [26] and Egress Program using TC [27]. It synchronizes the sequence number and the acknowledgment number between the two connections. Two BPF Maps (seq map, diff map) are used to transfer information between two programs.

Upon receiving the special SYN packet from the front-end server, the Ingress Program inserts the acknowledgment number (i.e., the sequence number the programmable switch replied) $L$ into the seq map using $\langle srcIP, srcPort \rangle$. The Web server using the kernel stack then returns a normal SYN-ACK packet. The Egress Program gets the number from the seq map, calculates the diff $\Delta$ between the sequence number $S$ of the SYN-ACK packet and the found number $L$ in the seq map, and then inserts $\Delta = S - L$ to the diff map. Since in our design, the LB does not modify any of the load content, the sequence numbers grow at the same rate for both connections, so the difference stays the same throughout the flow. For the subsequent ingress packet, Miresga adds $\Delta$ to the acknowledgment number, and for the subsequent egress packet, Miresga deletes $\Delta$ to the sequence number. When the ingress or egress receives the RST/FIN packet, the entry is removed from the BPF Map.

## 4.2 Lightweight Protocol Stack

In L7 load balancing, the critical task is to obtain and parse the application layer protocol, which is why the front-end server functions more as a bridge, connecting the client and the back-end server. Although utilizing the kernel protocol stack is convenient for deployment, its inherent complexity and redundancy are detrimental to Miresga's performance. To mitigate this issue, the front-end server employs our custom-designed lightweight protocol stack. In this subsection, we mainly show the details of the compressed TCP finite state machine (FSM) and the packet I/O. We take HTTP 1.0 as an example, while the discussion of other protocols is deferred to Appendix B.

**TCP FSM.** To handle stateful TCP protocol, we design a TCP FSM to handle the states in the two connections. The task of establishing connections with the client is simple and stateless, so we delegate it to the programmable switch. Considering the potential for SYN-Flooding attacks, we only pass the flow to the lightweight protocol stack for processing after the client sends a request. The front-end server stores the states and relevant information of all connections in a hash map called the Full Connection Table in the memory, using the client IP and client port, i.e., $\langle CIP, CPort \rangle$ as the key. The state transition diagram of the FSM is shown in Figure 4. The state is initialized to $INIT$. When Miresga receives the packet from the client with a new request from a new flow (①), Miresga parses the request, finds the corresponding load balancing rule, and sends

the special SYN packet whose acknowledgment number is set to the acknowledgment number of the cached packet minus 1 to the back-end server. After Miresga obtains the SYN-ACK response of the server (② and ②'), Miresga sends the cached HTTP GET packet to the back-end server. If the flow needs to be offloaded to the programmable switch, Miresga prepares the information required to be offloaded. The offloaded data will be translated into table entries and updated to the programmable switch, and the state will be set to *OFFLOADED*. Otherwise, the state will be *COMPLETE*. In states *COMPLETE* and *OFFLOADED*, the front-end server just performs connection splicing by modifying the IP and port, and then forwarding it. Especially, if Miresga detects a packet containing a new request that requires a different back-end server to process, it will terminate the current connection to the back-end server and establish a connection to the new back-end server. When the flow is completed, the front-end server returns the RST packet, removes the entry in the Full Connection Table, and informs the programmable switch to delete the corresponding entry if needed.

Retransmissions are inevitable throughout the flow. Since both the client and server need to acknowledge the correct receipt of data packets, retransmissions occur if no response is received from the other side within a certain period. Therefore, we have designed a passive retransmission mechanism. The implementation details are provided in Appendix C.

**Packet I/O.** For Miresga, as we do not require any kernel stack support, we utilize DPDK [36] to receive and send packets. By bypassing the kernel, DPDK enables us to achieve line-rate packet processing on the front-end server. We dedicate a core to each rx/tx queue to leverage multithreading capabilities.

### 4.3 State Consistency Maintenance

Similar to existing proxy-like approaches, Miresga acts as a forwarder, performing TCP splicing. It then sends packets to the client or server after the LB has established connections with both the client and the server. However, the stateful nature of the TCP protocol requires us to synchronize the state of the same flow across different components. As the front-end server is responsible for storing all flow states, Miresga must synchronize the states between the front-end server and the programmable switch, as well as between the front-end server and the back-end agent.

For the former task, there are two main subtasks to be addressed: obtaining the initial sequence number of the client and the programmable switch and synchronizing the state of the flows to be offloaded to the programmable switch. For the first one, the front-end server can retrieve the initial sequence number of the client and the programmable switch by subtracting 1 from the sequence number and the acknowledge number. For the second subtask, the state-of-the-art solution, Prism, blocks all traffic for a flow until its state is fully synchronized, a process that can introduce notable latency, particularly during the transmission of small objects. In contrast, Miresga's strategy does not require that the state on the programmable switch is always consistent with the front-end server. Miresga permits the programmable switch to temporarily deviate from the state of the front end. For flows involving small objects, the extra latency incurred by adding an entry to the switch often exceeds the latency savings from offloading the flow. Consequently, Miresga opts to offload only those flows where the duration of

subsequent packet delivery significantly surpasses the additional latency caused by offloading. For other flows, Miresga refrains from blocking traffic until the entry is written to the hardware, allowing the front-end server to handle packets in the interim. A potential connection issue arises when a single flow has multiple requests that need to be offloaded to the programmable switch and processed by different back-end servers. Before we finish the modification of the Offloaded Connection Table entries, the front-end server has broken the connection with the old back-end server. The data sent by the client will be directed to the old back-end server based on the outdated rules, and the old back-end server will return the RST, resulting in the connection closing abnormally. To prevent such occurrences, Miresga employs a two-step process: it immediately removes the entry upon receiving a new request and reinserts it following the reception of the SYN-ACK. This strategy provides Miresga with the necessary time to delete the relevant entries before the first packet from the client arrives, thereby precluding errors.

For the latter task, Prism both synchronizes the state and avoids the tedious TCP three-way handshake by passing serialized TCP information. However, this design means that we need to modify the Web server running on the back-end server, which complicates deployment. Given that the back-end agent only requires knowledge of the two initial sequence numbers to synchronize the sequence numbers and the acknowledgment numbers of the two connections, we can leverage the unused acknowledgment number field in the SYN packet to convey this information. The front-end server sends a special SYN packet whose acknowledgment number is set to the sequence number returned by the programmable switch, and the sequence number is set to the initial sequence number that the client chooses so that we do not need any extra overhead to synchronize information between the front-end server and the back-end server.

## 5 Implementation

We implemented the Miresga programmable switch on a 3.2 Tbps, 2-pipelined Intel Tofino programmable switch with a built-in Intel Xeon D-1527 8-core CPU and 16GB of memory. We implemented the Miresga programmable switch using ~1K lines of P4$_{16}$ code for the data plane and ~1.4K lines of C code for the control plane to receive the entries from the front-end server and insert them into the offloaded connection table. The two pipelines are the same as described in §4.1. We implemented the Miresga front-end server using ~1.5K lines of C++ code with DPDK-19.11. We implemented the Miresga back-end agent using ~0.5K lines of C code.

## 6 Evaluation

In this section, we focus on comparing the performance of existing software and hardware load balancing schemes with Miresga and assessing the performance improvement of Miresga. We intend to answer the following questions:

- Does Miresga outperform software and hardware load balancing designs in terms of end-to-end throughput (§6.2.1) and end-to-end latency (§6.2.2) across different response sizes?

- Is Miresga more suitable for application in heavy-tailed traffic distributions like those in data centers (§6.2.3)?

- Can Miresga scale to take on larger traffic (§6.3)?

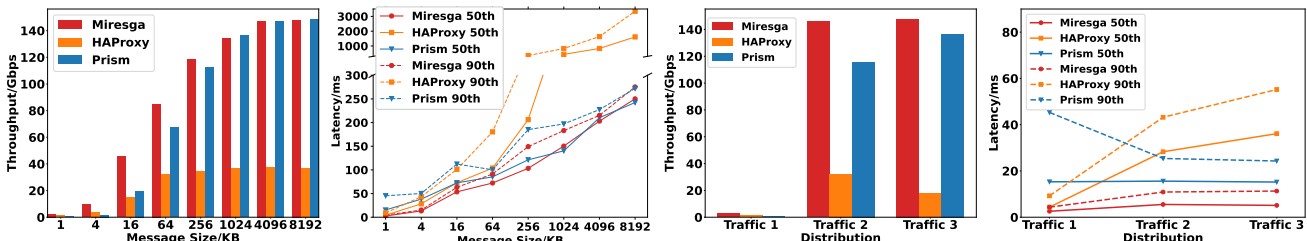

(a) Throughput comparison with different response sizes.  (b) Latency comparison with different response sizes.  (c) Throughput comparison with different traffic distributions.  (d) Latency comparison with different traffic distributions.

Figure 5: End-to-end throughput and latency comparison with different message sizes and using heavy-tailed traffic distributions. The distribution of the response size in Traffic 1 is 100% 1KB; Traffic 2 is 70% 1KB, 20% 10KB, 8% 1MB, and 2% 10MB. The distribution of the response size in Traffic 2 is 50% 1KB, 30% 10KB, 15% 1MB and 5% 10MB. We only measure the latency when the response size is 1KB.

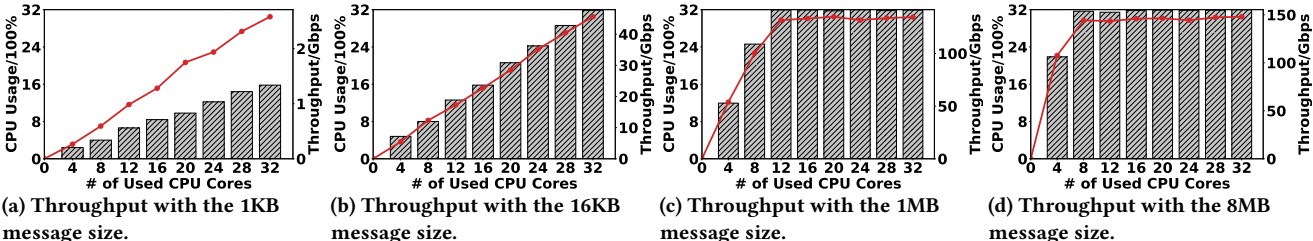

(a) Throughput with the 1KB message size.  (b) Throughput with the 16KB message size.  (c) Throughput with the 1MB message size.  (d) Throughput with the 8MB message size.

Figure 6: Throughput with different CPU core numbers used. The gray bar chart shows the CPU usage of the back-end server

- Can our table entry compression and back-end agent design effectively conserve resources (§6.4)?

## 6.1 Testbed

Our testbed consists of a 3.2 Tbps, 2-pipeline Intel Tofino programmable switch, 4 servers with 24-core CPUs and 64GB of memory, and 1 server with 32-core CPUs and 128GB of memory. Each server is equipped with a 100 Gbps Mellanox ConnectX-6 or 100 Gbps Intel E810CQDA2 NIC. We use two 16-core servers to generate HTTP requests and another two 16-core servers to handle the request. We use the 32-core server to run the software L7 LB/front-end server for comparison. We use Nginx as the back-end server to handle the requests. The programmable switch runs the Miresga/Prism program when testing Miresga/Prism and acts as a router when testing software load balancing.

**Baselines**. We choose HAProxy with f-stack [67] as our software-based comparison baseline because it is widely adopted and provides high performance. We implemented Prism [35] on the same programmable switch as our hardware-based comparison baseline.

## 6.2 Performance

*6.2.1 End-to-end Throughput.* We test the end-to-end throughput of Miresga, HAProxy, and Prism with different message sizes. We use *wrk* [33] to generate the HTTP requests and use Nginx as the Web server. We also use Lua scripts to modify HTTP headers to request different files, simulating traffic in an actual production environment. We perform L7 load balancing based on domain names, directing traffic with the domain name *mysite1.com* to one server, and traffic with the domain name *mysite2.com* to another server. Prior to this, we conducted a preliminary test to determine if offloading the flow could accelerate the delivery of subsequent packets. We observe that the performance gain from offloading becomes significant when the message size reaches 64KB, effectively offsetting

the latency incurred by the offloading procedure. Consequently, we have selected 64KB as the threshold for offloading.

We measure throughput by multiplying the number of requests per second (RPS) reported by *wrk* with the response size. We set the response sizes and then use *wrk* to issue the corresponding HTTP requests. Figure 5a shows the throughput comparison of Miresga, HAProxy, and Prism. In the case of large object transfers, Miresga offloads the data transmission process—constituting the majority of packets during the entire access—to the programmable switch. As a result, the throughput advantage of Miresga over HAProxy increases with the response size. At a message size of 64KB, the throughput of Miresga reaches 3× that of HAProxy, and when the response size is increased to 8MB, the throughput of Miresga reaches 4× that of HAProxy. For small message sizes, Miresga also achieves high performance due to its simplified protocol stack, with throughput that is still 2× better compared to HAProxy. For Prism, the hardware and software parallel strategy of Miresga makes it achieve 5× throughput than Prism for small objects. For large objects, we find that our scheme performs similarly to Prism, as both Miresga and Prism delegate the delivery of these objects to the programmable switch.

*6.2.2 End-to-end Latency.* We use the same method as in §6.2.1 to issue requests. We run tests for three minutes at each response size and use *tcpdump* [10] to capture the *pcap* files for analysis. Figure 5b shows the average and the 90th latency with different response sizes for the three designs. Since Miresga simplifies the TCP stack, uses DPDK to improve the processing speed for small objects, and leverages the programmable switch to accelerate the big message transmission, Miresga has less latency than HAProxy for both small and large objects. For Prism, it needs two entry operations per request and blocks the traffic before the state is fully synchronized, which leads to an increase in latency. The increase is

most obvious for small objects. Miresga circumvents this problem well because of its hardware and software parallel design. Although our overall design uses more components, the exception case of a flow is handled entirely by the client and the back-end server. This design makes our system more stable. The result that the latency of the 90th percentile is not much different from the average latency proves this. In contrast, HAProxy, where the load balancer manages two connections, is more susceptible to abnormal conditions. For Prism, frequent entry modifications lead to greater latency fluctuations when dealing with smaller objects. Consequently, Prism exhibits the opposite behavior for small objects: the difference between the 90th percentile latency and the average latency is more pronounced for smaller objects.

*6.2.3 Real-world Traffic Simulation.* In real data centers, traffic distribution is often heavy-tailed, where a small proportion of flows contribute to the majority of the data. To evaluate the performance of Miresaga under a real-world workload, we generate two types of heavy-tailed traffic distributions and use *wrk* to send requests according to these distributions to the back-end servers. Figure 5c shows the throughput comparison. For software load balancing, as latency increases, the corresponding RPS decreases, resulting in a decline in throughput. Since Prism does not perform well when dealing with small objects, its RPS is not as good as Miresga, resulting in some degradation in throughput. Figure 5d shows the latency of the packets whose response size is 1KB. As mentioned in §2.2, when using heavy-tailed traffic distributions, different flows will compete with each other for the limited CPU time, causing latency to increase. In contrast, Miresga offloads the elephant flows to the programmable switch with high performance, ensuring that elephant flows do not influence other flows processed by the front-end server. There is only a slight increase in processing latency compared to when all responses are 1KB. Prism, on the other hand, is limited by its slow handling of small objects, resulting in higher latency. With the increase of the proportion of large flows, the number of requests issued by the client per second decreases so that the pressure on the entries insertion becomes smaller. As a result, the latency of Prism on 1KB objects decreases. These two results indicate that Miresga is better suited for handling traffic distribution in data centers.

## 6.3 Scalability

In addition to throughput and latency, another metric to measure the performance of load balancing is scalability. We demonstrate that our system scales well by adjusting the number of CPU cores used by the front-end server. The result is shown in Figure 6. We empirically choose two small message sizes (1KB, 16KB) and two big message sizes (1MB, 8MB) to test. With small message sizes, the Miresga front-end server, which processes all packets, becomes the primary bottleneck. As the number of CPUs increases, the throughput rises linearly, and similarly, the CPU usage of the back-end servers also increases linearly. This demonstrates that we can enhance throughput further by deploying additional front-end servers. The scenario changes when dealing with larger objects. In this case, the front-end server can achieve very high throughput with minimal CPU resources, thanks to offloading the majority of packet delivery to the programmable switch. The bottleneck shifts to the back-end server capacity, which quickly reaches full utilization

| Method | Maximum # of Concurrent Connections |
|--------|-------------------------------------|
| **b+c** | 0.16M |
| **a+b** | 0.86M |
| **a+c** | 1.02M |
| **a+b+c** | 1.43M |

**a**: entry compression, **b**: table arrangement, **c**: back-end agent

**Table 1: The maximum number of concurrent connections with and without a specific optimization method.**

per CPU. Therefore, we believe that our system's performance can be significantly improved by increasing the number of back-end servers.

## 6.4 Maximum Number of Concurrent Connections

To prove that our designs effectively improve the table capacity, we successively test the largest table entry capacity before and after using each optimization: the table compression, the table arrangement, and the back-end agent. Table 1 shows the results. The entry compression greatly compresses the space of one entry and merges the two entries that were originally set for the inbound and outbound traffic distribution, thus providing the largest increase (∼9×). By properly scheduling the entries in ingress and egress, we allocated 2 additional stages for the offload connection table, resulting in a capacity increase of nearly 40%. Without the back-end agent, we need to store an additional 32-bit sequence number difference, which is 1.66× larger than without the back-end agent.

## 7 Limitations

Although we have attempted various optimization methods to enhance performance, our design still faces several limitations. Firstly, the speed at which the built-in CPU added the entries to the Offloaded Connection Table is low, making it challenging to increase the CPS. Secondly, our design processes encrypted packets more slowly because it is hard to perform the TLS termination on the programmable switches, so we need software help to do TLS termination. However, there are several hardware acceleration techniques available, so we believe that achieving TLS termination on programmable switches is not impossible. Finally, although we have compressed table entries, the limited size of SRAM still constrains performance.

## 8 Conclusion

In this paper, we propose Miresga, which, to our knowledge, is the first attempt to accelerate L7 load balancing using programmable switches. To leverage the full performance potential of the programmable switch, Miresga partitions the L7 load balancing process into three distinct parts: connection establishment, rule application, and data transmission. By offloading elephant flows to the programmable switch, which constitutes the bulk of the traffic, to the switch, Miresga achieves a significant boost in throughput and a reduction in latency. Our experiment proves that Miresga is more suitable for heavy-tailed traffic distributions, which are similar to those in the data center. In the future, we plan to expand our design further, such as by exploring the introduction of RDMA to expand the limited memory of programmable switches.

*This paper does not raise any ethical issues since all the traffic is simulated and does not contain any personal data.*

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

## A Table Arrangement

Since the same stage in the ingress pipeline and the egress pipeline sharing the resources and the limited SRAM prevents us from expanding the connection capacity. We specially allocated the tables

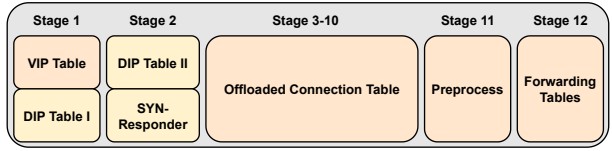

**Figure 7: Table Arrangement**

described above to make full use of the in-switch memory. Figure 7 shows the table arrangement of Miresga using a typical 12-stage programmable switch as an example. Since some tables do not occupy all the SRAM in a stage, our design integrates these tables and places them in a single stage. In the first two stages, we place VIP Table I, DIP Table I, and DIP Table II together. We arrange the Forwarding Tables at the last stage of the ingress pipeline. Since multiple cases arise in the Packet Steering module, preprocessing is needed and we place it in the second-to-last stage. The remaining stages are allocated to the Connection Query module. Through this design, we maximize the SRAM block size allocated for the Offloaded Connection Table.

## B Other Protocol Discussion

In this section, we discuss the differences of Miresga when using protocols other than HTTP 1.0.

Unlike HTTP 1.0, HTTP 1.1 [24] allows a single TCP connection to be reused for multiple requests, and these requests may need to be forwarded to different servers. In the programmable switch, Miresga can detect the presence of the content of the application layer in a packet by comparing the total length field in the IP header with the calculated header length obtained from the sum of the IP header length and the TCP header length. Therefore, when a packet with the application layer data is received from the client, the programmable switch will send the packet to the front-end server to check if the destination server needs to be changed. If the server to which the client is connected changes, Miresga simply closes the old connection, establishes a new connection with the new server, and then modifies the corresponding entry in the Offloaded Connection Table if needed.

To avoid head-of-line blocking, HTTP 2.0 [15] and SPDY [14] propose out-of-order delivery of responses. We can still forward the packets by adjusting the IP/TCP header correctly. However, our design does not fully support multiplexing, especially when we intend to route different requests to different servers, as we need to establish connections with multiple servers simultaneously. In fact, this poses a challenge for any design of the L7 load balancing because we can not support frames from multiple different streams within a single packet. Essentially, we are transmitting data across multiple connections, contradicting the principle of multiplexing, where multiple data transfers are completed within a single connection. Therefore, if we receive multiple requests at the same time, we will cache these requests. After the first request is completed, Miresga closes the old connection, establishes the new connection, and sends the next request. However, if all requests can be directed to a single server for processing, no modifications to the design are needed. The design of adjusting IP/TCP headers would still facilitate data migration.

QUIC [41] uses UDP rather than TCP as the transport layer protocol. However, QUIC encrypts its header so that we can't use the eBPF program to modify the header. We leave that for future work.

## C Retransmission

Although the lightweight protocol stack just does an IP/port modification after the state of the flow transmitting to *COMPLETE*

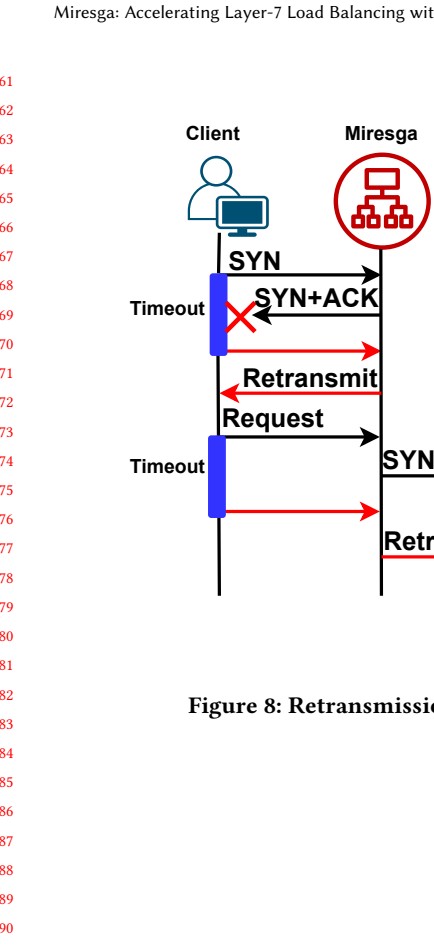

**Figure 8: Retransmission Example**

or *OFFLOAD*, it still needs to manage retransmissions before it. Miresga utilizes the retransmission mechanism of the client and server to handle possible packet loss. Miresga does not need to consider packet loss from the client or server side, as the client or server will automatically retransmit packets if no response is received within the timeout period. The Packet Processor will handle only the retransmitted packets. Thus, Miresga only needs to address the potential loss of packets sent by the front-end server or the programmable switch. Figure 8 illustrates how we handle retransmissions. Since we want to lighten the load of the front-end server, we prefer the front-end server not to actively decide whether to retransmit outgoing packets but rather to make this decision passively based on whether duplicate packets are received. Miresga stores the last received packet and the last sent packet of each flow. Whenever a packet is received, Miresga first checks if the received packet is the same as the last received packet. If it is, Miresga simply resends the last sent packet. Through this method, Miresga achieves a passive retransmission mechanism.

