# OpenReview forum: "Miresga: Accelerating Layer-7 Load Balancing with Programmable Switches"
_ACM.org/TheWebConf/2025/Conference — WWW 2025 Poster_

### Official Review · Reviewer_gSwr · 2024-11-20

**Novelty:** 3
**Technical Quality:** 3

**Review:**

Paper Summary:
This paper focuses on the optimization of Layer-7(L7) load balancing in cloud data centers, where a large amount of traffic needs to be properly distributed among the server clusters. To ensure the quality of service, the key challenge is how to enable the load balancers to achieve high throughput and low latency simultaneously. The authors propose a novel solution, dividing the tasks of load balancers, offloading the elephant flows to programmable switches and coordinating different components during the transmission. The paper’s contributions include using programmable switches to accelerate L7 load balancing, and extensive simulations for evaluation. In summary, the paper addresses the challenge of introducing programmable switches to L7 load balancing and improves the performance of load balancers.

Strengths:
1.	In order to enhance L7 load balancing, the paper introduces a novel hybrid system named Miresga that co-designs software and hardware. By offloading traffic-intensive tasks to programmable switches and leveraging general-purpose servers for complex protocol parsing, Miresga addresses the performance limitations within current load balancing solutions.
2.	The paper tackles significant challenges when utilizing programmable switches to help L7 load balancing, like limited memory and complex kernel protocol stack. By applying entry compression and designing task-specific agent, Miresga significantly leverages the advantages of programmable switches.
3.	The paper demonstrates technical depth by formulating the problem, dividing and conquering respectively. The technical rigor, detailed system design, explicit workflow and open-source code contribute to the paper's strength.

Weaknesses:
1.	The paper’s simulations show that Miresga is better than Prism in throughput and latency especially for small objects. When the message size becomes larger, the two approaches perform alike. The value of improvements for small objects should be discussed more to emphasize the advantages offered by Miresga.
2.	Although the paper touches on scalability by adjusting the number of CPU cores used by the front-end server, the programmable switches could be the bottleneck in extremely large or diverse network environments. How would Miresga keep its performance in different scales and scenarios?
3.	In Figure 6, the gray bar chart shows the CPU usage of the back-end server, and the max CPU usage is 32%. What is the reason that the CPU usage is bounded? Could we get over it simply by L7 load balancer design?
4.	The paper introduces programmable switches to accelerate L7 load balancing, but these switches are usually more expensive than commodity switches. Compared to traditional software methods, is the increase in cost acceptable?
5.	Since the system design processes encrypted packets more slowly, may attackers exploit the vulnerability to launch an attack against data center servers?
6.	The figures and charts in this paper could benefit from some improvements. Specifically, in Figure 1, the fonts used are a little bit small. In Figure 2 and Figure 3, the chart styles are rather complicated, and there is room for greater clarity in their presentation. Enhancing the quality and visual appeal of these visual elements would likely enhance the overall readability and impact of the paper.

**Questions:**

See weaknesses.

**Reviewer Confidence:**

3: The reviewer is confident but not certain that the evaluation is correct

**Scope:**

3: The work is somewhat relevant to the Web and to the track, and is of narrow interest to a sub-community

---

### Official Review · Reviewer_Guwd · 2024-11-25

**Novelty:** 5
**Technical Quality:** 6

**Review:**

- Strengths

1. This work proposes a joint software-hardware optimization approach, offloading L7 elephant flows to programmable switches while keeping mouse flows on general-purpose servers. It also optimizes kernel space performance to improve throughput. This is a timely contribution that highlights a promising direction for leveraging programmable switches.
2. The paper has a well-organized structure and strong readability. The design of the components is logical, closely aligned with the stated challenges, and achieves a good level of completeness.
3. The paper presents some interesting insights and includes a fair amount of detail, which provides better intuition for the proposed approach.

- Weaknesses

1. There is a lack of concrete data to support some claims, making it difficult to grasp the severity of the issues. For instance:
   * *"Moreover, there is competition for limited CPU resources among different flows."*
     How intense is this competition? What metrics are used to quantify it?
   This raises a concern: server CPU utilization in data centers is typically low. While it’s true that the OS network stack incurs overhead from a large volume of data requests, this contention is usually manageable. Previous studies by Microsoft and Alibaba have shown that CPU utilization is often below 40%, and servers dedicated to load balancing are even lower. Thus, I question whether the background of this challenge is fully justified.
2. The experimental evaluation lacks detailed insights. One of the key motivations of this work is to reduce CPU resource contention, utilizing tools like eBPF. However, the evaluation focuses primarily on E2E performance, without presenting low-level metrics that could provide a clearer picture of the performance improvements' origins. More details would help in understanding the benefit behind this work and evaluating the reliability of the proposed system.

**Questions:**

See  weaknesses.

**Reviewer Confidence:**

3: The reviewer is confident but not certain that the evaluation is correct

**Scope:**

4: The work is relevant to the Web and to the track, and is of broad interest to the community

---

### Official Review · Reviewer_GKSb · 2024-11-25

**Novelty:** 6
**Technical Quality:** 5

**Review:**

Pros
- The author has fully leveraged the advantages of programmable switches in handling elephant flows, pioneering the application of programmable switches to accelerate Layer 7 load balancing.
- The author's experimental section thoroughly demonstrates the superiority of the new system over traditional solutions, while also validating the effectiveness of the subsystem design.

Cons
- When the traffic on a connection shifts from a mouse flow to an elephant flow, the article does not describe how the system responds to the sudden change in traffic. Such a transition can impact the system's adaptability.
- Some typo and minor writing issues slightly affect the readability of the paper.

**Questions:**

- In Section 2.2, it is mentioned that the solution cost for equipping LB with SmartNICs is enormous. What specific economic advantages does Miresga have over the SmartNIC solution in terms of overall system expenditure?
- The article does not mention the specific methods for identifying elephant flows and mouse flows; please provide a detailed introduction of the methods used by the system. Additionally, if the traffic on a particular connection suddenly switches from a mouse flow to an elephant flow, does the system have the capability to reroute the processing path in response to sudden traffic surges?
- Does entry compression in programmable switches bring any negative impacts?
- Can the system be applied to load balancing on other layers?

**Reviewer Confidence:**

3: The reviewer is confident but not certain that the evaluation is correct

**Scope:**

4: The work is relevant to the Web and to the track, and is of broad interest to the community

---

### Official Review · Reviewer_6Yj5 · 2024-11-29

**Novelty:** 4
**Technical Quality:** 4

**Review:**

This paper presents Miresga, a new scheme for accelerating L7 load balancing using programmable switches. The solution divides the L7 load balancing process into three parts: connection establishment, rule application, and data transfer, and offloads elephant streams to programmable switches for high throughput and low latency. Experimental results demonstrate that Miresga performs better in handling heavy-tailed traffic distribution similar to that of a data center.

**Questions:**

1. What are the pressing issues with  layer-7 load balancing？ This is the key motivation for the authors’ work. Therefore, the authors should give a detailed introduction on problem description in Section 1.

2. In fact, traffic is usually bursty and the system is slow to add new table entries, does it not quite meet the real needs?

3. In subsection 6.2.3, the load is only for high tail delays, which is not very reasonable, is there a real load trace to experiment with in order to be convincing?

**Reviewer Confidence:**

3: The reviewer is confident but not certain that the evaluation is correct

**Scope:**

4: The work is relevant to the Web and to the track, and is of broad interest to the community

---

### Official Review · Reviewer_d542 · 2024-12-01

**Novelty:** 5
**Technical Quality:** 5

**Review:**

The authors developed Miresga, a tool for web load balancing that combines programmable switches and software solutions for speed and scalability at L7. Miresga divides the L7 load balancing task into three main subtasks of (1) connection establishment, (2) application protocol parsing, and (3) packet forwarding through connection splicing. These three tasks are coordinated between 3 processing components: (1) the programmable switch, (2) the front-end server and (3) the back-end server.

The system is then designed to coordinate between these elements to coordinate the tasks in a packet processing pipeline. The experiment results show improvement over existing stand-alone software and hardware solutions.

The key innovation here is the combination of the hardware programmable switch and the software modules and the orchestration between these components.

The work is innovative with major performance improvement over existing work. The paper is well written with a few minor presentation issues:
(1) "Offloaded Connection Tanle"
(2) Finish the explanation of  other steps (10-13) in the workflow example in Section 3.2
(3) VPort, CPort, DPort, etc are consistently used without first properly explaining what they are

**Questions:**

The authors offer very little discussion in the paper regarding the security aspect of Miresga. For example, how does MIresga deal with potential attacks such as  DDoS, SYN floods or HTTP floods on the programmable switch?

Furthermore, can Miresga be extended to handle connection authentication and access control mechanisms?

Miresga seems to be designed to be used for a single site. How does Miresga support global load balancing over geographically distributed networks, as well as multi-site or multi-cloud deployments?

**Reviewer Confidence:**

2: The reviewer is willing to defend the evaluation, but it is likely that the reviewer did not understand parts of the paper

**Scope:**

4: The work is relevant to the Web and to the track, and is of broad interest to the community